# Zambia: A Narrative Review of Success and Challenges in Lymphatic Filariasis Elimination

**DOI:** 10.3390/tropicalmed9010021

**Published:** 2024-01-15

**Authors:** Kingford Chimfwembe, Hugh Shirley, Natalie Baker, Richard Wamai

**Affiliations:** 1Department of Research and Postgraduate Studies, Chreso University, Lusaka 37178, Zambia; kchimfwembe@yahoo.com; 2Ministry of Health, Lusaka 10101, Zambia; 3Program in Medical Education, Harvard Medical School, Boston, MA 02115, USA; 4African Center for Community Investment in Health, Nginyang P.O. Box 48-30404, Kenya; r.wamai@northeastern.edu; 5Integrate Initiative for Global Health, Department of Cultures, Societies and Global Studies, College of Social Sciences and Humanities, Northeastern University, Boston, MA 02115, USA; 6Department of Global and Public Health, University of Nairobi, Nairobi 00100, Kenya; 7Nigerian Institute of Medical Research, Federal Ministry of Health, Lagos 101212, Nigeria

**Keywords:** neglected tropical disease, Sub-Saharan Africa, Southern Africa, elephantiasis, lymphatic filariasis, climate change

## Abstract

The establishment of the Global Programme to Eliminate Lymphatic Filariasis (GPELF) to stop the transmission of infection has significantly reduced the incidence of lymphatic filariasis, a debilitating mosquito-borne neglected tropical disease. The primary strategies that have been employed include mass drug administration (MDA) of anthelminthics and morbidity management and disability prevention (MMDP). While some countries have been able to reach elimination status in Africa, there is still active transmission of LF in Zambia. The nematode responsible for the disease is *Wuchereria bancrofti*, which is transmitted by *Anopheles* mosquitoes. To alleviate the suffering of those infected by the disease, the Zambian Ministry of Health launched a program to eliminate LF as a public health problem in 2003. This project reviewed the efforts to achieve the elimination of LF in Zambia, past and present government policies, and the anticipated challenges. MDAs have been conducted since 2014 and coverage has been between 87% and 92%. Zambia has now moved towards pre-transmission assessment surveys (PRETAS) and transmission assessment surveys (TAS). MMDP is a major priority and planned to be conducted between 2022 and 2026. COVID-19 presented a new challenge in the control of LF, while climate change, immigration, co-infections, and funding limitations will complicate further progress.

## 1. Introduction

The establishment of the Global Programme to Eliminate LF (GPELF) in 2000 by 2020 marked the beginning of one of the most ambitious global health programs to date [1]. Significant progress has been made to that end; of the 81 endemic countries at the turn of the century, 18 countries have achieved elimination of LF as a public health problem, defined as reducing the prevalence of LF in the community to levels below which transmission or re-emergence of disease is unlikely [2]. Bangladesh is the most recent success as of 2023 [3,4]. In 2018, the global population infected with LF was estimated at 51 million people across Africa, Southeast Asia, and the Americas, with the greatest burden thought to be in Southeast Asia [5]. This is a decline of approximately 25% since 2000, when an estimated 199 million people were infected with LF [5]. Despite this progress, over 880 million people live in areas where LF is still endemic [6]. In Sub-Saharan Africa, LF is endemic in over 30 countries, despite the expansion of programming to support elimination [6]. Thus far, Malawi and Togo are the only two countries in Sub-Saharan Africa that have successfully eliminated LF as a public health problem [6]. Among the remaining endemic countries, Zambia continues to be an important focus of the disease.

Zambia is a landlocked country in Southern Africa which gained independence in 1964. Zambia is made up of 10 provinces that are subdivided into 115 districts [7]. The demographics of Zambia show a young and growing country. The estimated population has increased to 19,610,769 in 2022 from 13,069,666 in 2010 [8] and there are slightly more females, at 10,007,713 (51%), compared to males, at 9,603,056 (49%). The rural population currently stands at 11,766,141 (60%) compared to the urban population of 7,844,628 (40%). Recent estimates show that poverty has declined in the country from 50% in 2016 to 44% in 2020. Poverty in rural areas declined from 69% in 2016 to 59% in 2020, while in urban areas, it declined from 25% to 18%. There is a link between poverty in a given country and the prevalence of neglected tropical diseases (NTDs) [9,10].

Of the 20 NTDs identified by the World Health Organization (WHO), 16 are known to be endemic in Zambia, one of these being LF [11,12]. LF is a vector-borne disease caused by *Wuchereria* and *Brugia* nematodes. The majority of cases are caused by *W. bancrofti,* including those infections that occur in Zambia [13]. *Anopheles*, *Culex*, and *Aedes* mosquitoes are known to spread LF, with *Anopheles* being the main vector in Zambia [14]. The clinical manifestations of LF occur secondary to chronic obstruction of the lymphatic vessels, resulting in debilitating lymphedema and classic elephantiasis, with the appearance of severe swelling of the lower extremities. The psychosocial effects of lymphatic filariasis on patients and their families, in addition to the health effects, further justify the importance of the global efforts to eliminate this disease [15,16,17]. In total, LF is estimated to result in the loss of 1.6 million disability-adjusted life years (DALYs) globally [18]. In men, scrotal hydroceles often develop due to lymphatic congestion; definitive management is hydrocelectomy, as percutaneous drainage results in only a temporary reduction in the size of the fluid collection.

The treatment for LF consists of a package of morbidity management and disability prevention (MMDP) consisting of antihelminth agents, surgical hydrocele management, and treatment for lymphoedema and adenolymphangitis [19]. The selection of a pharmacologic treatment for LF depends on co-endemic NTDs, namely loiasis and onchocerciasis, but generally consists of either monotherapy or some combination of albendazole, ivermectin, and diethylcarbamazine (DEC). Both ivermectin and DEC are associated with adverse reactions in patients with high loa loa parasite burdens, and ivermectin is contraindicated due to its association with severe encephalopathy in these patients [20]. Meanwhile, DEC is associated with a severe inflammatory response, called the Mazzotti reaction, in patients with onchocerciasis. Therefore, the determination of co-endemic diseases is important for determining which therapies are safe. Proper hygiene is also essential for managing lower extremity lymphedema and has been shown to reduce disease morbidity [21]. Therefore, comprehensive management of LF includes vector control, mass drug administration (MDA), access to diagnostic testing, hygiene, pharmacologic and surgical treatment, and strong surveillance programs.

Zambia is on the road to elimination, and, like other countries, has made significant strides. In this piece, we will review the efforts to achieve elimination of LF in Zambia with an eye toward the history of the disease in Zambia, past and present government policy, and future challenges.

## 2. Methods

A comprehensive literature search was conducted using PubMed, Google Scholar, and the World Health Organization (WHO) databases. The search included articles using the keywords “lymphatic filariasis”, “elephantiasis”, “Zambia”, “Sub-Saharan Africa”, and “neglected tropical disease.” Additional articles were acquired through an evaluation of articles cited by those from the initial search. Additional key documents were included outside of the search, including Zambian government documents and policies. The search was not year-restricted. Only articles in English were reviewed. The relevant articles were evaluated for thematic trends, while author and expert opinion and experience guided the structure of the review and identified gaps in our knowledge and the current research.

## 3. Lymphatic Filariasis in Zambia

The first suspected cases of LF in Zambia were reported in the 1930s around Luangwa in Lusaka Province [22]. Building on this information, further LF surveys were undertaken in 1970 by Barclay in the Luangwa basin [23]. The first definite cases of LF due to *W. bancrofti* were reported by Hira between 1975 and 1977 [24,25,26]. Then, 20 years later, case reports identified *W. bancrofti* in blood smears from patients from both Southern and Northen provinces [27,28].

Lymphatic filariasis (LF) is endemic in 96 of the total 116 districts in the country [8]. Prevalence ranges from 7.4% to 54% across the country, with the greatest burden by percentage found in Lusaka and Western Provinces [29]. Prevalence of LF is determined by detecting the circulating filarial antigen (Ag) produced by the *W. bancrofti* parasite. Samples that test positive for the antigen have additional testing, with a thick blood smear to measure the microfilaria (Mf) counts [30]. The vector for LF in Zambia, just like in most Southern African countries, is the *Anopheles* mosquito, which is also the principal vector for malaria in the country and in the region [29,31,32]. *Anopheles funestus* and *An. gambiae* are predominant in Eastern, Luapula, Northern, and Lusaka provinces, while *An. arabiensis* is restricted to the southern region [14,33,34,35].

Local beliefs about LF influence health-seeking behavior, especially as they relate to MMDP. Debilitating conditions, such as lymphedema and hydroceles, are likely underreported both in Zambia and globally, due, in part, to stigma surrounding these conditions. In Zambia, absolute numbers of these conditions have been reported in the hundreds; however, the true number is likely far greater [7]. Qualitative interviews performed in Zambia’s Luangwa region found that many people often perceived LF as a hereditary disease or one that is spread through sexual contact or by contact with animal feces [36].

## 4. Historical Government Policies

The road to eliminating LF started in 1993, when LF was identified as probably eradicable by the International Task Force for Disease Eradication at the CDC using contemporary technologies [37]. In 1997, resolution 29 of the 50th World Health Assembly spotlighted LF as a target for elimination [38]. This was quickly followed by the official launch of the GPELF in 2000 as a collaborative effort between governments, pharmaceutical companies, non-profit organizations, and other stakeholders to achieve the elimination of LF by 2020. In response, the Zambian Ministry of Health launched a program to eliminate LF as a public health problem in 2003 [29]. The strategies of this call to action were to interrupt transmission through a program of MDAs and to alleviate the suffering of clinically affected populations using MMDP. This began with mapping the distribution of the disease across the country, which was completed in 2011 using rapid immunochromatographic test cards [29]. Zambia was subsequently included for the first time in the annual WHO GPELF progress report for the 2012 update (Table 1) [39]. The 2013–2017 NTD strategic plan was the first government policy document outlining and aligning national NTD priorities with the international goals and milestones. This strategic plan was reworked into the NTD Master Plan 2015–2020, alongside the establishment of an NTD unit in the Ministry of Health. Zambia has since published two more NTD Master Plans for the 2019–2023 and, most recently, 2022–2026 periods [7,40].

The results of the prevalence mapping conducted for LF between 2003 and 2011 indicated an average national prevalence of 7.4%, with a range of 0–54% [29]. LF endemicity was recorded in 96 of the 116 districts in all the 10 provinces, with 95% of the population placed at risk. Due to the endemic status of the country, MDAs were instituted in all endemic districts [7,29,40]. The MDA exercise was conducted following WHO guidelines using albendazole and DEC as the drugs of choice because loiasis is not known to be co-endemic with LF in Zambia. Onchocerciasis is also not known to be endemic; however, the disease is endemic in neighboring countries, namely Angola, the Democratic Republic of the Congo, and Mozambique, and mapping studies are underway to better understand the burden of this disease in Zambia [8,41,42].

The first province to receive the MDAs was Western province in 2013 since the highest prevalence of LF was recorded in Kalabo, a border town with Angola [7,43]. MDAs for other provinces commenced in 2015 through to between 2020 and 2021 [8]. MDA coverage for LF in the country has been between 87% and 92% and approximately 12 million people have been treated during each round of MDA. MDAs are conducted using both door-to-door and distribution point methods with volunteer community health workers [7]. MDAs are ideally conducted annually, with an independent coverage survey between each round to assess programme coverage. These data are reported to GPELF and are summarized annually alongside LF data from other countries participating in MDAs. The results from Zambia are recorded in Table 1.

Between 2018 and 2019, funding challenges limited the scope of Zambia’s MDA implementation. In 2018, this meant reducing the number of implementation units (IUs), which are the smallest geographic units which MDAs are organized around, with the number of active MDAs reduced to 69 from the total of 85. Within those 69 IUs, 100% of IUs met the threshold of effective coverage, defined by treating greater than 65% of the population within the IU. In 2019, funding limitations precluded any MDA in Zambia.

**Table 1 tropicalmed-09-00021-t001:** Annual data from Zambia on progress of mass drug administrations (MDAs) since data reporting was included in the annual Global Programme to End Lymphatic Filariasis (GPELF) reports.

Year	Pop. Requiring MDA (mil.)	Total IUs	IUs with MDA	IUs MDA Achieving >65% Coverage (%)	Geographical Coverage (%)	Pop. in Active MDAs (mil.)	Pop. Treated by MDA (mil.)	Program Coverage (%)	National Coverage (%)
2015	11.6	85	83	92.8	97.65	11.6	10.7	92%	92%
2016	11.3	85	85	97.7	100	11.3	10.4	92.10%	92%
2017	11.3	85	-	-	-	-	-	-	-
2018	12.0	85	69	100	100	10.9	11.2	102%	93%
2019	12.0	85	-	-	-	-	-	-	-
2020	13.2	85	35	97.1	41.2	4.64	4.88	105%	37%
2021	13.2	85	35	97.1	41.2	4.64	4.88	105%	37%

This includes the total population requiring MDA based on the population of known endemic areas and number of implementation units (IUs), defined as the smallest unit at which decisions regarding MDA implementation are made, often the district level, and the number of IUs with active MDAs. The percent of IUs achieving affective coverage is estimated based on a target of >65% coverage of the population within the IU. The total population targeted by active MDAs and the total number of people treated is used to calculate the program coverage and gives a sense of how successful MDAs are in delivering drugs to their catchments. The geographic coverage is calculated by dividing the IUs implementing MDAs over the total number of IUs and gives a sense of the proportion of IUs that requires MDAs or is actively undergoing MDAs each year. Finally, the national coverage is defined as the total population treated over the population requiring MDA and gives a sense, at the national scale, of what proportion of the at-risk population is covered by an MDA. While Zambia was included in the 2012–2014 updates, limited data are available; therefore, these years have been excluded from the table. MDA interruptions due to COVID-19 and funding challenges account for the decline in active MDAs while improvements in survey techniques and a growing population account for the rising population requiring MDAs. In 2019, funding limitations precluded any MDAs. There are no data available for 2017. Data are compiled from annual GPELF updates [44,45,46,47,48,49,50].

## 5. The COVID-19 Era and Beyond

In the post 2020 period, government policies on LF are outlined in the 2022–2026 NTD Master Plan [40]. The plan blueprints the Ministry of Health’s vision to transform Zambia into a nation free from preventable diseases in accordance with the SDGs, Vision 2030, Zambia National Health Strategic Plan, and the 8th National Development Plan [8]. It sets out two primary LF targets: to reduce the DALYs related to LF by 75 percent and to eliminate LF by 2026. The strategies outlined to achieve this are largely similar to those from prior NTD master plans and include MDAs; MMDP; surveillance; water, sanitation, and hygiene (WASH) programs; and increasing community awareness. WASH programs are important for LF because proper cleaning and hygiene improve and prevent the progression of lymphedema [21]. Educating patients and nurses using a package of care that includes hygiene with water and soap is therefore essential for LF MMDP.

The WHO roadmap to LF elimination as a public health problem progresses through the stages of baseline mapping, MDA, pre-transmission assessment surveys (PRETAS), transmission assessment surveys (TAS), surveillance, and verification. In areas where LF prevalence at baseline exceeds a 1% threshold, WHO recommends that an MDA is conducted in all implementation units for at least five rounds with a minimum coverage goal of at least 65%. This is followed by PRETAS—a process of sentinel and spot check site assessments in each IU to determine whether the prevalence in each site is less than 1% based on Mf tests or less than 2% for Ag. Sentinel and spot check sites have the same population dynamics as those being sampled, except that a sentinel site remains the same during the course of the program while a spot check site may change dynamically. LF control programs that satisfy the requirements for PRETAS then move to TAS, which aim to determine whether LF prevalence is below 1% for Mf or less than 2% for Ag through a more systematic survey of six- and seven-year-old children in all endemic communities. Three TAS surveys are carried out, spaced two years apart to determine whether there is active transmission in endemic areas and whether MDAs may still be required. After TAS, LF control programs move into surveillance, where active transmission is monitored five years after MDA. The last stage in the LF elimination pathway is verification. National control programs use both historical and epidemiological data to confirm that there is no active transmission of the disease in a designated area [2,44].

Between 2020 and 2021, Zambia successfully conducted the fifth and final round of MDA as per the WHO guidelines [1,51]. The country has moved toward pre-transmission assessment surveys (PRETAS) and transmission assessment surveys (TAS) and may move into LF surveillance and vector control activities. Between 2021 and 2022, PRETAS activities were conducted in 80 (83%) of the 96 endemic districts, while the remaining PRETAS for the 16 districts will be conducted in 2023. TAS, LF passive surveillance, and LF vector control activities are expected to be conducted between 2023 and 2026. As Zambia transitions from active MDAs to ongoing surveillance, the role of MMDP in LF management will become increasingly important. This is based on the number of 1696 hydroceles and 1871 lymphoedema cases identified between 2015 and 2021 for which surgeries have not yet been performed [40]. The next step in ensuring that the surgeries are conducted for hydroceles and lymphedemas is training for surgeons, nurses, community health workers, patients, and family members to LF patients [52]. Coordination between local health facilities and NTD program managers is essential to ensure patient care is safely and equitably handed off. A robust monitoring mechanism is required so that progress can be recorded for all the surgeries conducted over a period of time.

The LF elimination program has been a success in Zambia despite some challenges. The country has been able to conduct all five rounds of MDA in all the 96 endemic districts as recommended by the WHO. In these MDAs, the average coverage has been 90 percent, significantly above the 65% WHO threshold. Recent data on the LF prevalence in Zambia suggest that there is a decreasing disease burden in the country. A circulating LF antigen prevalence of 0% has been recorded in some settings in Zambia [53] demonstrated that LF prevalence in Zambia significantly reduced from 11.6 per cent in the period 2003–2005 to 0.6 per cent between 2012 and 2014.This significant decrease has been attributed to MDA and successful malaria programs through insecticide treated nets and indoor residual spraying.

The vectors for LF and malaria in Zambia are *Anopheles* mosquitoes [29,31,32]. The LF vector control program, though it has not yet commenced, has already benefitted from an intensive vector control program under malaria. In Zambia, malaria control is under the National Malaria Elimination Programme (NMEP). The NMEP, along with other stakeholders, has conducted malaria vector control in the country using indoor residual spraying (IRS) and with the use of insecticide-treated nets (ITNs). This synergic approach to the control of malaria has significantly reduced its prevalence from 17 per cent in 2015 to 9 per cent in 2018 [54,55]. This has also coincided with a significant decrease in LF prevalence, as reported by Nsakashalo-Senkwe et al. [53]. In provinces where the malaria burden remains high, such as Muchinga, Luapula, and Central provinces, the intensification of LF screening programs has been proposed. Because the mosquito vector for malaria and LF is the same in these areas, the failure of malaria control programs creates the opportunity for LF recrudescence. Therefore, an integrated approach to malaria and LF control in areas still recording high malaria prevalence is proposed [56,57].

## 6. Ongoing and Future Challenges

Despite the progress made in Zambia over the last two decades, numerous challenges remain or have arisen that will make reaching the goal of elimination exponentially more difficult. Climate change is and will continue to disrupt many facets of life across Zambia, including due to its impacts on the epidemiology of vector-borne diseases [8,58]. Additionally, the key remaining challenges include accessing difficult-to-reach locales, obtaining adequate national and international funding, treating co-infections, and maintaining programmatic resilience in the face of systemic disruptions such as COVID-19.

### 6.1. Climate Change

Climate change is positioned to impact the prevalence and distribution of LF through altered vector and transmission dynamics [59]. These mechanisms include changes in temperature and precipitation, which are key parameters in defining the geospatial distribution of LF across which the disease can occur [60]. More frequent or intense rainfall creates more stagnant water, optimal for LF vectors like *Aedes*, *Anopheles*, and *Culex* mosquitoes, and longer/warmer summer seasons may extend suitable weather for vector breeding and transmission. However, the precise impacts require complex models for their accurate prediction due to the nuances of the environmental factors at play. For example, in terms of the duration of the LF season, a drier, hotter spring with a warmer, wetter fall is predicted to lead to a net delay in the breeding season of *C. quinquefasciatus* and increase the vector distribution into western Zambia [41]. Slater and Michael also found a nonlinear relationship between rainfall and the probability of LF prevalence, hypothesizing that while an increase in rainfall may provide an increased suitable habitat in some areas, too much rainfall may start to wash away egg-laying sites [60]. LF occurrence peaks were also seen between 25 C and 32.5 C, and an increase in winter temperatures had a dramatic effect on LF occurrence [60]. Even so, taking the effect of these nuances into account, their model predicts that the population at risk in Africa could increase from 543–804 million in 2012 to 1.65–1.86 billion by 2050, with the largest effects in Zambia, Zimbabwe, and Angola [60]. Given LF prevalence has been found to depend heavily on population, climate migration and its impacts on agricultural practice may also affect exposure if populations move out of more heavily impacted areas such as the Southern province [60,61,62]. In addition to climate-driven migration, an influx of political and economic migrants from nearby countries, such as the Democratic Republic of the Congo, Rwanda, Burundi, Angola, and Somalia, may also alter the prevalence of LF and other NTDs in Zambia, LF-inclusive [63].

Essential to LF programming is the monitoring of the prevalence and burden of LF. Just as people migrate to more hospitable environments, the vectors of LF will adapt to climate change according to migration. This poses a key challenge to making progress in eliminating LF, as maintaining an agile surveillance system that accounts for changes in vector-habitable areas will be important. This is so that a reduction in the prevalence of LF in a historically endemic area is not taken as a sign of complete success, given the vector may have moved to historically non-endemic, and therefore unmonitored, areas. Climate models that predict where disease vectors are likely to be prevalent are important for ensuring that resources are allocated to new potential disease foci so that treatment and monitoring systems can be in place prior to the arrival of the disease.

In sum, while the effects of climate change will be heterogeneous, models predict a net increase in transmission risk in light of the climate crisis, and distributional maps which capture this complexity may be the best tools for the GPELF to plan for the future. 

### 6.2. Health Systems

Co-infections with malaria, which is transmitted by the same vectors as LF, can worsen LF clinical manifestations and disease progression. Because helminth infections can suppress immune responses, co-infection is thought to increase morbidity and mortality [64]. Additionally, LF treatment recommendations change in the setting of co-infections such as onchocerciasis or loiasis [65]. As discussed, mapping on onchocerciasis is underway in Zambia given the prevalence of this disease in bordering countries. Climate change will likely impact the distribution of both malaria and onchocerciasis, just as it will affect LF. Studies into the effects of climate change on onchocerciasis epidemiology are scarcer, but one study demonstrated an inverse parabolic relationship between heat and black fly prevalence [66]. As temperatures in more equatorial countries exceed the ideal, we may see co-infections of onchocerciasis and LF in Zambia. As such, many have advocated for the closer, albeit nimble, integration of LF and other tropical diseases [52,67].

Even before the COVID-19 pandemic, funding, logistical, and organizational challenges have led to discordance between the aspirational goals and achieved outcomes of MDAs. LF programs in Zambia have encountered budgeting challenges secondary to operating in a political landscape with fluctuating funding for NTDs, an obstacle also shared by MDA programs in Guinea and Nigeria [68,69]. LF elimination programs are consequently dependent on foundations, philanthropy, and bilateral or multilateral funding partners. While these axillary sources of funding were important for gaining initial momentum, the third WHO report on NTDs estimated that a significant portion of required funding is not likely to be sourced from foreign donors in the future [11,70].

From a logistical standpoint, coordination between government entities such as the NTD Unit at Zambia’s Ministry of Health and the National Malaria Elimination Centre could also be improved. For example, the data on vectors that have been collected by the National Malaria Elimination Centre could help improve planning within the NTD Unit at the Zambian Ministry of Health [71]. Another logistical challenge has been the heterogenous uptake of vector control measures such as bed nets. The data support that certain provinces like Luapula in Zambia continue to display a high malaria burden despite years of control measures, indicating continued vulnerability to LF secondary to mosquito exposure [72]. Without intervention, the discrepancy between the program goals and execution, secondary to funding and logistical challenges, can pose challenges for program monitoring and evaluation, hampering the ability to track progress, and iterate on and identify areas for improvement for the future.

The COVID-19 pandemic has exacerbated many of the above challenges, disrupting healthcare systems, straining public health programs, and limiting individual patients’ access to care. Mandated social distancing and travel restrictions likely made access to diagnosis and treatment for LF more difficult to access, leading to delays in care and likely increased morbidity. At the systems level, as a neglected tropical disease, resources were diverted from LF initiatives, including mass drug administration, vector control efforts, and community-based interventions. Delayed programs or temporary suspension of such programs has led to gaps in LF control [73,74,75]. In addition, limitations on travel and gathering mean that data collection on LF surveillance programs also was likely limited, especially in more remote or underserved areas [76].

## 7. Conclusions

Great strides have been made toward the elimination of LF in Zambia since the turn of the millennium. Many challenges remain for the country as it moves from the active MDA phase toward post MDA surveillance. Namely, the prevalence of co-infections, such as malaria, HIV, and the possibility of co-endemic onchocerciasis, pose roadblocks to progress. At the same time, they create new opportunities for collaboration across disease-focused working groups through synergistic strategies and the pooling of increasingly limited funds.

Other challenges in opportunities for change and improvement include the decentralization of program implementation, improving the in-country capacity for training and research, and optimizing the current systems for malaria to improve LF monitoring. Decentralizing program implementation from the Ministry of Health to the level of local health facilities may improve the efficiency of resource allocation. Currently, the Ministry of Health dictates day-to-day activities rather than providing supervisory expertise. Funding for LF programming going toward training courses on LF diagnostic skills using grassroots medical training will sustain the program beyond donor life. Capacity building for LF control can be strengthened by the involvement of academic institutions in the country that can specifically train LF experts. The LF control program can also set up an active surveillance that can ride on the malaria screening program where every sample tested for malaria can also be tested for LF, depending on the feasibility of some health facilities setting up such a system. While district-specific control maps have been used in the past, health-facility-specific maps may give a clear picture on the status of elimination of the disease in the country, and spatial tools such as a geographic information system and remote sensing can be used to develop vector-based models for LF transmission. As the country moves into a new NTD Master Plan, LF must remain top of mind in order to ensure the elimination of this ancient and terrible disease from Zambia.

## Data Availability

No new data were created or analyzed in this study. Data sharing is not applicable to this article.

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
