# Peer review of "Zambia: A Narrative Review of Success and Challenges in Lymphatic Filariasis Elimination"

_tropicalmed, 2024, doi:10.3390/tropicalmed9010021_

Round 1
Reviewer 1 Report
Comments and Suggestions for Authors
Tropicalmed-2605629
The review article is a manuscript of great relevance, as it correlates the topics with facts chronologically that occurred over the years, which point out the critical points and successes. And Lymphatic Filariasis is a disease that has a low mortality rate, but a high rate of comorbidity. Thus causing much damage to the affected population. Therefore, it is important to emphasize that measures need to be taken for its control. Within what was reported in this manuscript, I missed the questioning regarding vector control. I suggest adding a vector control and monitoring topic.
Abstract: ok
Keywords: failed to describe
1. Introdutinon
-Line 32: The reference "1" was missing in the text
-Between line 58 and 67: The reference "18" is missing in the text
2. Lymphatic Filariasis in Zambia
-Between line 87 and 98: The reference "16; 31 and 34" is missing in the text
5.1 Climate Change
-Line 210: where it says wCulex modify to Culex and italicize vector genera “Aedes; Anopheles; Culex”
-Between line 219 and 226: The reference "53" is missing in the text
5.2 Health Systems
-Between line 272 and 284: The reference "67" is missing in the text
Author Response
Thank you to Reviewer 1 for their thoughtful comments and suggestions. The attached word document is a table with responses to each point and relevant lines in the updated manuscript.

Reviewer 2 Report
Comments and Suggestions for Authors
The main issue with this manuscript is the lack of innovation and originality. The strategies discussed in the manuscript are too general and do not provide any new insights into the elimination of lymphatic filariasis. The authors have not provided any new or innovative strategies that could be implemented to eliminate lymphatic filariasis in Zambia. The manuscript is a mere summary of the current state of lymphatic filariasis elimination in Zambia, and it does not provide any new information or insights. Additionally, the references cited in the manuscript are not up to date, which suggests that the authors have not conducted a thorough literature review. The references cited in the manuscript are outdated, and there are no recent studies cited in the manuscript. The authors should conduct a more thorough literature review and update their references to include more recent studies. Furthermore, the table is not well-organized, and the data presented in it are not helpful. The manuscript lacks practical strategies that could be implemented to eliminate lymphatic filariasis in Zambia. The authors have not provided any practical strategies that could be implemented to eliminate lymphatic filariasis in Zambia. The manuscript is a mere summary of the current state of lymphatic filariasis elimination in Zambia, and it does not provide any practical strategies that could be implemented to eliminate lymphatic filariasis in Zambia. The study appears to be more of a report than a research article. The manuscript is not well-structured, and it does not follow the standard structure of a research article. The manuscript is more of a report than a research article, and it lacks the necessary components of a research article. Overall, the manuscript is too simple to be called a paper. Therefore, I cannot recommend this manuscript for publication in its current form. The manuscript should be restructured to follow the standard structure of a research article.
Comments on the Quality of English LanguageN/A
Author Response
Thank you to Reviewer 2 for their thoughtful comments and suggestions. The attached word document is a table with responses to each point and relevant lines in the updated manuscript.

Reviewer 3 Report
Comments and Suggestions for Authors
This was an interesting article but could be improved. Specifically, in several places terms and/or abbreviations are not fully explained. For example, NTD is used in one paragraph but is not defined until the next paragraph.
Line 93: how is prevalence determined?
Line 131: you should explain why the presence of Loa Loa and onchocerciasis affects drug choice.
Line 146: what is an implementation unit?
Line 166: the WASH program is mentioned only in passing. More detail is needed.
Line 188: in several places malaria is capitalized.
Line 68: use the term disability adjusted life but not DALY, use DALY later in manuscript.
Please explain how PRETAS, TAS and passive surveillance are done.
The table is hard to understand and terms and abbreviations are not clearly explained. Tables should be able to stand alone.
Additional descriptive information is needed in many places so that a general audience will be better able to understand.
Comments on the Quality of English Language
The sentence structure is complex and makes the article difficult to understand at times.
Author Response
Thank you to Reviewer 3 for their thoughtful comments and suggestions. The attached word document is a table with responses to each point and relevant lines in the updated manuscript.

Reviewer 4 Report
Comments and Suggestions for Authors
This manuscript is timely and well written. The introduction captures the essence of the Global Programme to Eliminate LF and clearly outlines the tools required to document transmission interruption. Other sections in the manuscript could not provide evidence for impact on transmission intensity and infectious prevalence. This is crucial for a review of successes. The requirements for pre-TAS surveys are based on number of successive MDAs that achieved epidemiological coverage. This review has demonstrated that Zambia has met the conditions to start pre-TAS but not success based on reduction in transmission intensity. That will be determined by the pre-TAS and TAS results. The sections on COVID-19 and climate change could be more relevant when we have information on impact.
I believe that Zambia is on track to achieve transmission interruption of LF because of the long history of vector control targeting malaria and human African trypanosomiasis (HAT). The following papers have strongly predicted this optimistic outcome.
1. Significant decline in lymphatic filariasis associated with nationwide scale-up of insecticide-treated nets in Zambia: 10.1016/j.parepi.2017.08.001
2. Reducing the population requiring interventions against lymphatic filariasis in Africa: https://www.thelancet.com/action/showPdf?pii=S2214-109X%2815%2900292-2
3. The importance of vector control for the control and elimination of vector-borne diseases: https://doi.org/10.1371/journal.pntd.0007831
I would like to suggest the following for a more appropriate review:
That the title be changed to ‘ Is Zambia on track to achieve LF elimination by 2030?’
The authors can focus the literature search on the impact of vector control using DDT and bednets on other diseases like malaria and HAT which are more difficult to control than LF. Following the arguments in the papers suggested above the authors can show that LF transmission was already on the decline before MDA was started and increase in bednet coverage in recent times could have impacted LF transmission as was demonstrated in Kenya and The Gambia ( see papers below).
1. Elimination of Lymphatic Filariasis in The Gambia: 10.1371/journal.pntd.0003642
2. Sustained reduction in prevalence of lymphatic filariasis infection in spite of missed rounds of mass drug administration in an area under mosquito nets for malaria control: 10.1186/1756-3305-4-90
Author Response
Greetings,
The authors would like to thank reviewer 4 for their time and energy spent improving our manuscript. Please find the attached document with our responses to each point raised by reviewer 4.

Round 2
Reviewer 2 Report
Comments and Suggestions for Authors
The authors do not address my concerns at all. No improvement is detected in the revised manuscript. I do not believe that the narrative review is suitable for journal publication. Therefore, I recommend to reject the manuscript.
Author Response
We appreciate the time Reviewer 2 has dedicated to providing feedback on the manuscript. Our response to round 2 is included in the attached document.

Reviewer 3 Report
Comments and Suggestions for Authors
The authors did a commendable job of responding to reviewer comments which greatly improved the manuscript.
Comments on the Quality of English LanguageThe English was fine.
Author Response
We appreciate the time reviewer 3 dedicated to providing feedback on our manuscript. We have responded to their comments in the attached document.

Reviewer 4 Report
Comments and Suggestions for Authors
The main queries have been addressed